# Enhancing Response Rates in Web-Based Surveys: The Impact of Direct Participant Contact

**DOI:** 10.3390/healthcare12141439

**Published:** 2024-07-19

**Authors:** Mélanie Suppan, Laurent Suppan, Tal Sarah Beckmann, Caroline Flora Samer, Georges Louis Savoldelli

**Affiliations:** 1Division of Anaesthesiology, Department of Anaesthesiology, Clinical Pharmacology, Intensive Care and Emergency Medicine, Geneva University Hospitals and Faculty of Medicine, 1205 Geneva, Switzerland; tal.beckmann@hug.ch (T.S.B.); georges.savoldelli@hcuge.ch (G.L.S.); 2Division of Emergency Medicine, Department of Anaesthesiology, Clinical Pharmacology, Intensive Care and Emergency Medicine, Geneva University Hospitals and Faculty of Medicine, 1205 Geneva, Switzerland; laurent.suppan@hug.ch; 3Division of Clinical Pharmacology and Toxicology, Department of Anaesthesiology, Clinical Pharmacology, Intensive Care and Emergency Medicine, Geneva University Hospitals and Faculty of Medicine, 1205 Geneva, Switzerland; caroline.samer@hug.ch

**Keywords:** web-based survey, survey administration, participation rate, individual tokens

## Abstract

Achieving a high participation rate is a common challenge in healthcare research based on web-based surveys. A study on local anesthetic systemic toxicity awareness and usage among medical practitioners at two Swiss university hospitals encountered resistance in obtaining personal email addresses from Heads of Departments. Participants were therefore divided into two groups: those who were directly invited via email (personal invitation group) and those who received a generic link through intermediaries (generic link group). The latter group was eventually excluded from survey data analysis. To determine whether one method of survey administration was more effective than another, we carried out a retrospective analysis of response rates and the proportion of new questionnaires completed after initial invitation and subsequent reminders. The results showed significantly higher response rates in the personal invitation group (40.2%, 313/779) compared to the generic link group (25.3%, 22/87), emphasizing the effectiveness of personal invitations on response rate (+14.9%, *p* = 0.007). The personal invitation group consistently yielded a higher number of completed questionnaires following the initial invitation and each reminder. The method of survey administration can greatly influence response rates and should be acknowledged as a quality criterion when conducting web-based surveys.

## 1. Introduction

Web-based surveys have become instrumental in data collection across various domains, notably in medical research. The widespread availability of internet access and digital platforms has made web surveys not only convenient but also highly advantageous. They enable swift administration and data acquisition at a reasonable cost while facilitating access to a wide-ranging participant pool. However, it is crucial to recognize their limitations, such as susceptibility to selection bias and the challenge of securing high response rates [1,2].

Several guidelines are available to assist researchers in developing and reporting studies carried out through web-based surveys, prioritizing high-quality standards and the use of evidence-based frameworks [3,4,5]. These guidelines advocate for thorough detailing of key survey components, such as survey administration, which is crucial for enabling reproducibility. Nevertheless, these guidelines do not offer definitive guidance on the comparative effectiveness of various approaches.

The method of survey administration itself significantly impacts participation rates, with web-based surveys typically yielding 11–12% lower rates compared to other survey modalities [6]. Various factors spanning the development, delivery, completion and return processes of web-based surveys can further influence response rates [7]. However, detailed descriptions of the initial email contact and subsequent reminders with potential participants are often lacking. Two common options include (1) direct access by the research team to the email addresses of potential participants for direct communication; (2) reliance on a third party to send the invitation and reminders. Reluctance to disclose personal email addresses, stemming from concerns about potential abuse or inappropriate dissemination, poses a common challenge in web-based survey research. Such concerns, which persist despite the presence of rather stringent data protection rules, often result in survey link transmission through an intermediary. This approach probably impacts participation rates, thus creating a selection bias, and could subsequently influence the validity and reliability of research findings.

To assess the knowledge of local anesthetic systemic toxicity (LAST) and the utilization of local anesthetics (LAs) among physicians across various medical specialties in two Swiss university hospitals, a web-based survey was conducted between 5 September and 2 November 2022 [8]. To enhance engagement and secure accurate email addresses, endorsement was sought from the Medical Heads of Departments of the relevant specialties beforehand. Although two Heads of Departments declined to provide individual email addresses, they allowed their secretaries to distribute a generic link to the study platform. Since this was a clear deviation from the study protocol, the responses obtained through such invitations were eventually excluded from this initial analysis. Based on the hypothesis that direct access to participants could enhance response rates, the present study is a secondary analysis of previously collected data. The aim of this analysis was to compare response rates between participants contacted directly by the research team and those reached through an intermediary.

## 2. Materials and Methods

### 2.1. Study Design and Setting

This was a retrospective analysis of prospectively collected data. A declaration of no objection was previously issued by the regional research ethics committee (Req-2021-00467), as the initial project fell outside the scope of the Swiss Act on Research involving Human Beings [9]. Given that this particular project was carried out on the same dataset, no additional submission was presented to the regional ethics committee.

The initial closed web-based survey was carried out and reported according to the Checklist for Reporting Results of Internet E-Surveys (CHERRIES) [5]. The survey was administered using a custom Joomla! 3.10 platform (Open Source Matters Inc., New York, NY, USA) hosted on a Swiss server (https://survey.anesth.ch, (accessed on 2 November 2022)). The Community Surveys component (version 5.9; Shondalai, BulaSikku Technologies Pvt, Hyderabad, Telangana, India) was used to create the web-based survey, with responses automatically recorded in an encrypted MySQL-compatible database (MariaDB version 10.3; MariaDB Corporation Ab, MariaDB Foundation, Middletown, DE, USA). AcyMailing 7.9 (Acyba, Lyon, France) was used to manage email distribution lists.

The 6-page questionnaire integrated a branching logic strategy, allowing for a condensed questionnaire based on participants’ answers. Prior to deployment, the questions underwent validation by a panel of experts, including a clinical pharmacologist and senior anesthesiologists. A comprehensive description of the questionnaire and the web-based platform has been previously outlined [8].

### 2.2. Participants

Medical practitioners belonging to relevant medical specialties and working in two Swiss university hospitals (Hôpitaux Universitaires de Genève—HUG, Geneva; and Centre Hospitalier Universitaire Vaudois—CHUV, Lausanne) represented the target population. The Medical Heads of the anesthesiology department and of all surgical specialties were asked to endorse the project prior to study inception. Those who agreed provided the email addresses of the physicians working in their department (17/29, 58.6%). After curation, these addresses were added to an AcyMailing distribution list. Two of the Medical Heads of Departments who refused to give us access to individual email addresses nevertheless agreed to let their secretaries send a generic link (to the study platform) to the physicians working in their departments.

The following two groups were therefore identified:The personal invitation group: participants contacted directly by the research team whose email addresses were added to the distribution list. These participants each received an invitation containing unique survey links which were automatically generated and contained individual tokens. The use of these unique links prevented double entries and enabled targeted reminders.The generic link group: participants whose individual email addresses were unknown to the research team. The exact same email with a generic link to the study platform (instead of a unique survey link) was sent to the secretaries of the concerned departments with a request to forward this email to the medical team.

Emails inviting physicians to participate in the study were sent on 5 September 2022. The invitation email stated the purpose of the study and acknowledged its approval by the relevant Head of Department. The estimated time for completion, the name and contact of the principal investigator and a disclaimer containing a data policy statement were displayed both on the invitation email and on the welcome page of the survey. Informed consent was gathered electronically. Participation was voluntary, and no incentive was given to promote participation. Four reminders were sent during the study periods to encourage participation in the non-responding population (20 September, 3 October, 17 October and 27 October). Targeted reminders and requests to the secretaries to send generic reminders were sent on the same date. The study website was put offline on 2 November 2022, thereby preventing any participation beyond this point.

### 2.3. Outcomes

The primary outcome was the participation rate in each group. The secondary outcomes were the proportion of newly completed questionnaires in each group after the first invitation and after each reminder.

### 2.4. Data Extraction and Statistical Analysis

Data were extracted to a Comma-Separated Value (CSV) file and imported for curation in Stata (version 17; StataCorp LLC., College Station, TX, USA).

Descriptive characteristics were reported using the median (Q1:Q3). The frequencies of categorical variables were calculated and reported in percents (%). The Mann–Whitney test was used to compare continuous variables, while Fisher’s exact test was applied to compare binomial or categorical ones. There was no need for imputation since the methods used to gather data prevented the presence of missing values. *p* values < 0.05 were considered significant.

## 3. Results

Throughout the study period, a total of 335 questionnaires were completed. In the personal invitation group, the participation rate was 40.2% (313/779), whereas in the generic link group, it was 25.3% (22/87). This marked a significantly higher response rate in the personal invitation group (+14.9%, *p* = 0.007).

There were no significant differences between groups regarding age, years of clinical experience or gender (Table 1). The response rate remained consistently higher in the personal invitation group following the initial invitation and subsequent reminders. Notably, no new questionnaires were completed in the generic link group after the second and fourth reminder, and only one was filled after the third reminder.

There was no significant difference measured when comparing the number of completed questionnaires after the initial invitation in the personal invitation group to the total number of completed questionnaires in the generic link group throughout the entire study period.

Figure 1 illustrates the study flow chart. Figure 2 provides a histogram plot detailing the proportion of accounts created during each study period for both groups.

## 4. Discussion

### 4.1. Main Considerations

This analysis reveals that employing personal invitations in web-based surveys yields a higher response rate compared to using intermediaries to disseminate a generic survey link. The observed increase in participation appears to be primarily attributed to the ability to issue targeted reminders rather than the distribution method itself. An increase of nearly 15% in the response rate is substantial for a study utilizing a web-based survey, particularly within the context of healthcare research where survey response rates tend to be notably low [10]. In a context where numerous factors can significantly influence the response rate [11], it becomes imperative to pinpoint practices that guarantee the highest possible response rate.

In a previous study, it was suspected that the low participation rates in a web-based study could be largely attributed to the failure of intermediaries to disseminate the study link [12]. This assertion seems logical, given that sharing a link imposes an additional workload on intermediaries. One plausible hypothesis is that intermediaries, who are often not directly involved in the study or invested in its outcomes, may refrain from distributing the link to avoid increasing their ongoing tasks. Supporting evidence for this hypothesis includes the complete absence of new completed questionnaires in the group provided with the generic link after the second and fourth reminders. Notably, there was only one new response following the third reminder. There is also a possibility that altering the original message may dilute its meaning, thereby diminishing interest and discouraging potential participation [13,14]. While one argument for circulating the link internally was to demonstrate the clear support from the Head of Department, it appears insufficient to outweigh the drawbacks associated with this approach.

In medical research, direct access to certain professionals or specific populations may be restricted due to gatekeeping practices aimed at safeguarding personal information and preventing unsolicited harassment. However, these practices pose significant barriers to accessing the target populations, necessitating consideration of alternative approaches. Regarding the general population, adhering to regulations such as the General Data Protection Regulation (GDPR) in Europe [15], or equivalent regulations elsewhere, is imperative. Nonetheless, providing intermediaries with details about the investigators, study protocols and evidence of training in good clinical practice should facilitate direct access to healthcare professionals for medical research purposes.

Furthermore, direct participant access enables better overall survey control. Utilizing personal tokens helps prevent duplicate entries and enables the interruption and resumption of surveys at the same point later on. Moreover, it facilitates the delivery of targeted reminders, thereby avoiding unnecessary outreach to individuals who have already participated.

### 4.2. Limitations

This study is subject to several limitations. First, its retrospective design poses a constraint. Originally, the study did not anticipate encountering significant challenges in accessing personal emails or analyzing the impact of distribution methods on participation rates. Consequently, responses collected through the generic link were excluded from the analysis to maintain fidelity to the initial protocol. Secondly, the findings are confined to a small cohort of participants from two university centers in Switzerland. Nevertheless, the recurring issue of distributing a generic link through intermediaries persists in our practice and consistently hampers participation. Finally, the lack of control and access to the generic link group not only complicates the identification of problems and understanding of difficulties but also has the potential to obscure other underlying issues.

### 4.3. Perspectives

Web-based surveys continue to offer a rapid and engaging method for asynchronously accessing populations and collecting research data. However, the ongoing challenge lies in increasing participation to ensure the gathered data can be considered representative and generalizable. Notably, in the healthcare sector, acquiring personal email addresses to generate personal tokens appears to be particularly challenging yet significantly enhances participation rates. Researchers would probably be faced with the same barrier when dealing with other categories of participants, and particular care should be taken to avoid selection bias. Addressing these obstacles is paramount, underscoring the crucial need for practices that facilitate direct access to potential participants for research purposes. Additionally, guidelines and recommendations should integrate these specifications and recognize them as essential quality criteria for conducting web-based surveys.

## 5. Conclusions

This study demonstrated that directly inviting and reminding participants to answer a web-based survey resulted in a substantial increase in the response rate. To attain response rates that yield representative and generalizable results, it appears preferable to refrain from relying on intermediaries to disseminate survey links. On the other hand, efforts should be made to facilitate the research team’s access to the participants’ email addresses.

## Figures and Tables

**Figure 1 healthcare-12-01439-f001:**
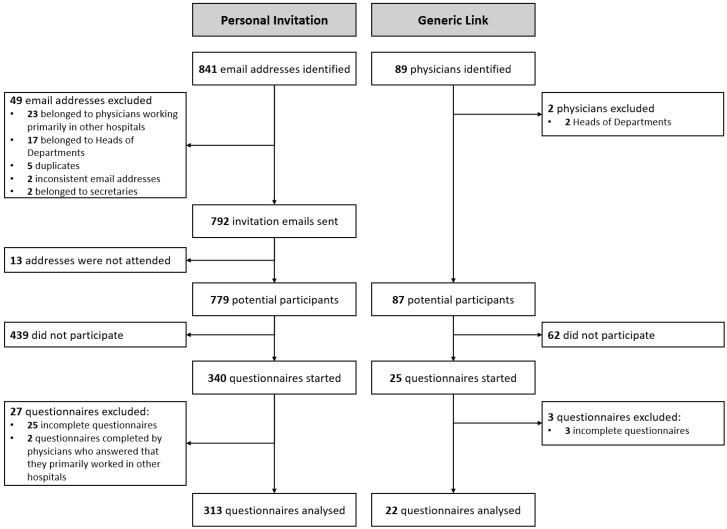
Study flowchart.

**Figure 2 healthcare-12-01439-f002:**
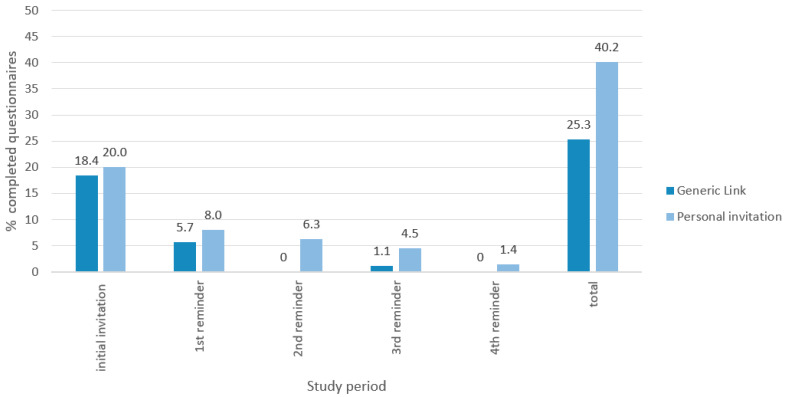
Proportion of accounts created during each study period.

**Table 1 healthcare-12-01439-t001:** Associations between participants’ characteristics and participation.

Characteristics	Personal Invitation Group (N = 313)	Generic Link Group (N = 22)	*p*-Value
Age (median, Q1:Q3)	35 (32:40)	35 (32:38)	0.671
Years of experience (median, Q1:Q3)	6 (4:11)	9 (5:10)	0.343
Gender (n, %)			
Man	170 (54.3%)	12 (54.5%)	>0.99
Woman	142 (45.4%)	10 (45.5%)	
Other	1 (0.3%)	0 (0%)	

## Data Availability

Data can be obtained from the corresponding author upon reasonable request.

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
