# Peer review of "Enhancing Response Rates in Web-Based Surveys: The Impact of Direct Participant Contact"

_healthcare, 2024, doi:10.3390/healthcare12141439_

Round 1
Reviewer 1 Report
Comments and Suggestions for Authors
This manuscript is well written. I have only one concern on the analysis. Given that the study was not designed to directly compare the response rate between the two distribution methods, this could introduce bias from confounding factors. Perhaps, performing a logistic regression could adress these confounding factors.
Author Response
Comment: This manuscript is well written. I have only one concern on the analysis. Given that the study was not designed to directly compare the response rate between the two distribution methods, this could introduce bias from confounding factors. Perhaps, performing a logistic regression could adress these confounding factors.
Response: Thank you very much for your review and for this comment. You are of course right, and we have therefore aknowledged the risk of bias under "4.2. Limitations". We have also attentively considered your suggestion of carrying out a logistic regression. In line with the first comment submitted by Reviewer 3, we ultimately decided to report only non-parametric tests. Therefore, and because of the lack of data regarding several potential confounding factors (particularly among the population of non-responders), we decided to refrain from performing a logistic regression.
Reviewer 2 Report
Comments and Suggestions for Authors
Thank you for the opportunity to review this paper.
When first reading the paper (without the abstract), I found it difficult to realize that this secondary analysis was initiated by the finding that not all survey participants of the initial study could be reached by their individual email address, thus leading to two groups that could be compared with regard to the method of contact (personal vs. generic link). It became clear only in the limitations section where the authors point to the difficulties in obtaining individual contact information of all eligible participants.
Moreover, the authors write in the introduction that data of participants who were contacted via a generic link were not included in the analysis (p. 2, lines 65, 66). This seems, however, to refer to the initial study, not this secondary analysis. In that respect, the abstract is clear and easy to understand whereas this is less clear with the manuscript.
I would therefore recommend that the authors revise the respective passages in the paper and make them clearer and less confusing.
Some minor suggestions:
p. 2, lines 53-54: Data protection legislation/rules might also be mentioned here as an influencing factor.
p. 2, lines 92 et seq.: Please specify the relevant/selected specialties included in the survey.
p. 6, lines 206 et seq.: It would also be interesting to know what challenges and benefits the authors see in web-based surveys with patients as opposed to healthcare professionals.
p. 7, references: Please add the missing information for reference 15 (General Data Protection Regulation).
Author Response
Major Comment: When first reading the paper (without the abstract), I found it difficult to realize that this secondary analysis was initiated by the finding that not all survey participants of the initial study could be reached by their individual email address, thus leading to two groups that could be compared with regard to the method of contact (personal vs. generic link). It became clear only in the limitations section where the authors point to the difficulties in obtaining individual contact information of all eligible participants.
Moreover, the authors write in the introduction that data of participants who were contacted via a generic link were not included in the analysis (p. 2, lines 65, 66). This seems, however, to refer to the initial study, not this secondary analysis. In that respect, the abstract is clear and easy to understand whereas this is less clear with the manuscript. I would therefore recommend that the authors revise the respective passages in the paper and make them clearer and less confusing.
Response: You are absolutely right: the fact that this was a secondary analysis was not easy to determine based on the version of the introduction you have reviewed. The final part of the introduction now reads: "Since this was a clear deviation from the study protocol, the responses obtained through such invitations were eventually excluded from this initial analysis. Based on the hy-pothesis that direct access to participants could enhance response rates, the present study is a secondary analysis of previously collected data. The aim of this analysis was to com-pare response rates between participants contacted directly by the research team and those reached through an intermediary."
Minor Comment 1: p. 2, lines 53-54: Data protection legislation/rules might also be mentioned here as an influencing factor.
Response: Thank you for this comment. The section has been updated thus: "Reluctance to disclose personal email addresses, stemming from concerns about potential abuse or inappropriate dissemination, poses a common challenge in web-based survey research. Such concerns, which persist despite the presence of rather stringent data pro-tection rules, often result in survey link transmission through an intermediary."
Minor Comment 2: p. 2, lines 92 et seq.: Please specify the relevant/selected specialties included in the survey.
Response: Your comment made us realize that more data was needed regarding the target population. Actually, all surgical specialties were initially targeted. Since we believe that it would have been too long and of limited yield to list all the surgical specialties represented in a university hospital, the text was adapted thus: "The Medical Heads of the anesthesiology department and of all surgical specialties". Further details are of course available in the original paper (Ref. 8)
Minor Comment 3: p. 6, lines 206 et seq.: It would also be interesting to know what challenges and benefits the authors see in web-based surveys with patients as opposed to healthcare professionals.
Response: This is indeed interesting - the following sentence has been added to highlight this issue: "Researchers would probably be faced with the same barrier when dealing with other cat-egories of participants, and particular care should be taken to avoid selection bias."
Minor Comment 4: p. 7, references: Please add the missing information for reference 15 (General Data Protection Regulation).
Response: Reference 15 has been updated accordingly: "European Union. Regulation (EU) 2016/679 of the European Parliament and of the Council of 27 April 2016 on the Protection of Natural Persons with Regard to the Processing of Personal Data and on the Free Movement of Such Data, and Repealing Directive 95/46/EC (General Data Protection Regulation). 2016. Available online: https://eur-lex.europa.eu/eli/reg/2016/679/oj (accessed on 5 May 2024)."
Reviewer 3 Report
Comments and Suggestions for Authors
Congratulations to the authors and thanks for their work.
The manuscript anticipates that having personal e-mail accounts and not relying on generic accounts and intermediaries increases the response rate in web-based questionnaires among a cohort of physicians.
The idea is exciting as it remedies the deficiency of a significant sample when we need to go to physicians and only have generic accounts. It establishes the idea of having a database of personalized accounts, which can be monitored with tokens and allows us to improve the responses received, as well as participation. I consider it a good starting point to begin studies and test different methodologies of form acquisition.
The introduction is correct and presents the subject of the study. The results, discussion, limitations, and conclusions align with what is presented in the paper. It is perhaps very concise but must be adapted to the context of a brief paper.
Some doubts and improvements that I will pass on to the authors for their explanation and consideration if necessary:
- In the methodology, they specify that parametric tests are used due to the sample size. Ideally, tests should have been performed to check whether they meet the assumptions of normality, and if not, non-parametric tests should be used.
- On the other hand, it would be interesting to add "the frequencies of those categorical variables are calculated", within the descriptive data methodology.
- It is specified that "there were no significant differences between groups in terms of age (p=0.671), sex (p=0.965) or years of experience (p=0.169)”, it would be interesting to show these data, either in a table or at least their mean or grouping.
- What was the reason for using an unpaired or independent test if the cohort is the same and it is a typical example of paired data? I imagine that these tests were done to compare the different collaborative responses after the 4 reminders. There is a lack of indication on this point.
- For the final response if independent testing is contemplated, as a final experiment and obviating the 4 reminders.
Regards.
Author Response
General Comment: Congratulations to the authors and thanks for their work. The manuscript anticipates that having personal e-mail accounts and not relying on generic accounts and intermediaries increases the response rate in web-based questionnaires among a cohort of physicians. The idea is exciting as it remedies the deficiency of a significant sample when we need to go to physicians and only have generic accounts. It establishes the idea of having a database of personalized accounts, which can be monitored with tokens and allows us to improve the responses received, as well as participation. I consider it a good starting point to begin studies and test different methodologies of form acquisition. The introduction is correct and presents the subject of the study. The results, discussion, limitations, and conclusions align with what is presented in the paper. It is perhaps very concise but must be adapted to the context of a brief paper. Some doubts and improvements that I will pass on to the authors for their explanation and consideration if necessary: [...]
Response: Thank you very much for this most encouraging comment and for your thorough and thoughtful review of our brief report. We have updated our manuscript in line with your specific comments.
Comment 1: In the methodology, they specify that parametric tests are used due to the sample size. Ideally, tests should have been performed to check whether they meet the assumptions of normality, and if not, non-parametric tests should be used.
Response: You are of course perfectly right. We first considered that the central limit theorem should apply given our sample size, but we agree that non-parametric tests may be more appropriate for our study given the very limited number of participants in the "generic link" group. We have revised the methods and results section of our manuscript accordingly.
Comment 2: On the other hand, it would be interesting to add "the frequencies of those categorical variables are calculated", within the descriptive data methodology.
Response: We apologize for this omission - this is now reported in the methods ("The frequencies of categorical variables were calculated and reported in percents (%)")
Comment 3: It is specified that "there were no significant differences between groups in terms of age (p=0.671), sex (p=0.965) or years of experience (p=0.169)”, it would be interesting to show these data, either in a table or at least their mean or grouping.
Response: Thank you for this suggestion: Table 1 has now been added and reports the detailed results computed according to the revised statistical analysis strategy.
Comment 4: What was the reason for using an unpaired or independent test if the cohort is the same and it is a typical example of paired data? I imagine that these tests were done to compare the different collaborative responses after the 4 reminders. There is a lack of indication on this point.
Response: We decided to perform an independent test since each questionnaire would have been started by a different participant, but this is of course debatable. This point is however already addressed by virtue of the new statistical analysis strategy prompted by your first comment.
Comment 5: For the final response if independent testing is contemplated, as a final experiment and obviating the 4 reminders.
Response: We were not certain to understand this last point but believe it should have been addressed through the change in the statistical analysis strategy.
Round 2
Reviewer 2 Report
Comments and Suggestions for Authors
Thank you for revising your manuscript. I think it is now easier to understand. My comments have been adequately taken into account.